# Socioeconomic inequalities, psychosocial stressors at work and physician-diagnosed depression: Time-to-event mediation analysis in the presence of time-varying confounders

Ana Paula Bruno Pena-Gralle[1,2,3]*, Denis Talbot[1,2], Xavier Trudel[1,2], Alain Milot[1,2], Mahée Gilbert-Ouimet[1,4], Mathilde Lavigne-Robichaud[1,2], Ruth Ndjaboué[5], Alain Lesage[6], Sophie Lauzier[7], Michel Vézina[1,8], Johannes Siegrist[9], Chantal Brisson[1,2,3]

1 Centre hospitalier universitaire (CHU) de Québec Research Center, Population Health and Optimal Health Practices Unit, Québec, Québec, Canada, 2 Faculty of Medicine, Laval University, Québec, Québec, Canada, 3 VITAM – Centre de Recherche en Santé Durable, Québec, Québec, Canada, 4 Department of Health Sciences, Université du Québec à Rimouski, Lévis, Québec, Canada, 5 School of Social Work, Sherbrooke University, Sherbrooke, Québec, Canada, 6 Centre de recherche de l'Institut universitaire en santé mentale de Montréal, Montréal, Québec, Canada, 7 Faculty of Pharmacy, Laval University, Québec, Québec, Canada, 8 Institut National de Santé Publique du Québec, Québec, Québec, Canada, 9 Faculty of Medicine, Department of Medical Sociology, Heinrich-Heine-University, Duesseldorf, Germany

* ana.paula.bruno.pena.gralle@umontreal.ca

## Abstract

### Objectives

There is evidence that both low socioeconomic status (SES) and psychosocial stressors at work (PSW) increase risk of depression, but prospective studies on the contribution of PSW to the socioeconomic gradient of depression are still limited.

### Methods

Using a prospective cohort of Quebec white-collar workers (n = 9188 participants, 50% women), we estimated randomized interventional analogues of the natural direct effect of SES indicators at baseline (education level, household income, occupation type and a combined measure) and of their natural indirect effects mediated through PSW (job strain and effort-reward imbalance (ERI) measured at the follow-up in 1999–2001) on incident physician-diagnosed depression.

### Results

During 3 years of follow-up, we identified 469 new cases (women: 33.1 per 1000 person-years; men: 16.8). Mainly in men, low SES was a risk factor for depression [education: hazard ratio 1.72 (1.08–2.73); family income: 1.67 (1.04–2.67); occupational type: 2.13 (1.08–4.19)]. In the entire population, exposure to psychosocial stressors at work was associated with increased risk of depression [job strain: 1.42 (1.14–1.78); effort-reward imbalance (ERI) 1.73 (1.41–2.12)]. The estimated indirect effects of socioeconomic indicators on depression mediated through job strain ranged from 1.01 (0.99–1.03) to 1.04 (0.98–1.10),

**Data Availability Statement:** Data cannot be shared publicly because of privacy and ethical restrictions. Data for the PROspective Québec cohort are available from the CHU de Québec-Université Laval Ethics Review Board (ethiquedelarecherche@chudequebec.ca) for researchers who meet the criteria for access to confidential data. Administrative data collected by the Régie de l'Assurance Maladie du Québec are available from the "Commission d'accès à l'information du Gouvernement du Québec" (https://www.cai.gouv.qc.ca) for researchers who meet the criteria for access to confidential data.

**Funding:** This work was supported by the Canadian Institutes of Health Research, https://cihr-irsc.gc.ca/e/193.html, Canada [Grant MOP-133542] to CB and by VITAM (Sustainable Health Research Center, https://vitam.ulaval.ca/), Canada to APBPG. DT was supported by a career award from the Fonds de Recherche du Québec – Santé (https://frq.gouv.qc.ca/sante/). There was no additional external funding received for this study. The funders had no role in study design, data collection and analysis, decision to publish, or preparation of the manuscript.

**Competing interests:** The authors have declared that no competing interests exist.

4–15% of total effects, and for low reward from 1.02 (1.00–1.03) to 1.06 (1.01–1.11), 10–15% of total effects.

## Discussion

Our study suggests that PSW only slightly mediate the socioeconomic gradient of depression, but that socioeconomic inequalities, especially among men, and PSW both increase the incidence of depression.

## Introduction

Socioeconomic inequalities often translate into health inequalities. Socioeconomic status (SES), commonly measured by indicators such as education, income, and occupation type [1], is a powerful predictor of health outcomes. When compared to socioeconomically privileged people, those with low SES have higher rates of mortality, morbidity and disability including higher rates of mental health problem [2]. While the etiology of depression is complex, its incidence has been associated with psychosocial and socioeconomic determinants [3]. Meta-analyses have reported the highest risks of depression among those with lower levels of education and those with lower income [4,5]. Importantly, a monotonic association is well supported, with those in the middle of the social gradient being also at increased risk, compared to the most privileged ones [2,6].

Socioeconomic inequalities are also present in the working environment. Workers of lower SES tend to be more exposed to psychosocial stressors, such as high psychological job demands, low decision latitude, high effort and low reward [7–9]. These stressors, in turn, are associated with the onset of several physical and mental health problems [10–12], including depression [13–16].

The contributions of psychosocial stressors at work to the socioeconomic gradient of depression have been explored in two previous prospective studies. While effort-reward imbalance (ERI) might explain in part the increased depression observed for people at low SES [17], the contribution from high psychological job demands and low job control is still uncertain [18]. Furthermore, despite the considerable difference between men and women in the prevalence of depression [3], the relation of psychosocial stressors at work with the socioeconomic gradient of depression has not been investigated for men and women separately.

The present study uses a prospective cohort of white-collar workers, psychosocial stressors at work from two widely used models, physician-diagnosed depression, and a design with temporal ordering of the exposure, mediator and outcome. The objectives are (i) to investigate the effects of SES indicators and of psychosocial stressors at work on the incidence of physician-diagnosed depression for men and women; (ii) to evaluate if psychosocial stressors at work mediate the effect of socioeconomic inequalities on the incidence of physician-diagnosed depression for men and women.

## Methods

### Population

We used questionnaire, interview and clinical data from PROspective Quebec (PROQ), a prospective cohort that has followed 9188 white-collar workers since 1991 in Quebec, Canada [19].

For the present study, three indicators of SES were measured at the first data collection wave ($T_1$, 1991–1993), and psychosocial stressors at work according to two recognized models at the second wave ($T_2$, 1999–2001). The sample was restricted to participants who consented to having their data linked with administrative data on physician services and hospital discharge collected by the provincial universal health provider *Régie de l'Assurance Maladie du Québec* (RAMQ) to identify physician-diagnosed depression following $T_2$ (see S1 Fig). All data were fully anonymized before we accessed them, and the study was approved by the ethical review board of the CHU de Quebec (2015–2063—F9-44103).

The exclusion criteria were: 1) death before $T_2$ (n = 117); 2) physician-diagnosed depression during the year before $T_2$ (n = 369); 3) not actively working at $T_2$ (n = 1174); 4) lack of linkage with administrative data (n = 387); 5) failure to participate in $T_2$ (n = 620). The analytical sample for the main analyses included 6521 participants (see S1 Fig).

## Socioeconomic status indicators

Three indicators of SES were measured using self-reported questionnaires, namely education, household income and occupation type. Education was assessed as the highest academic classification: less than completed college, at least 2 years college (14–17 years), and 2 years college + completed university ($\geq$ 17 years); household income was categorized as less than CAD40 000, CAD40 000 to CAD69 999, or at least CAD70 000 before taxes, roughly corresponding to revenues below, around and above the mean national household income of CAD 44,783 [20], respectively; occupation type was categorized as managers, professionals, and others (administrative staff, technicians, and manual workers).

To create a combined measure of SES, we used principal component analysis, a technique that takes a set of several related factors and extracts from them the combination of weights that describes the common information most parsimoniously [21] (S1 Table). This combined factor was divided into tertiles corresponding to "low", "medium" and "high" SES.

## Psychosocial stressors at work

Psychosocial stressors at work were measured according to the Demand-Control (DC) and ERI models as dichotomous variables. Using the validated French version of the Karasek questionnaire with a four-point Likert scale, scores for psychological job demands (9 items, Cronbach's $\alpha$ 0.73) and job control (9 items, $\alpha$ = 0.81) were calculated at $T_1$ and $T_2$ by summing the items [7,22]. Job strain was considered present when psychological job demands (range 9–36) were at least 24, the median of a representative sample of Quebec workers, while job control (range 24–96) was at most 72 [23]. Job strain was considered absent in all other cases.

Reward was measured by 9 original items from the validated French version of the Siegrist ERI questionnaire with a four-point Likert scale [8,24]. Since the Siegrist effort scale was not measured in our cohort, we used psychological job demands as a proxy [25]. Imbalance was considered present when the ratio of the sum of scores for demands, divided by the sum of scores for reward, was > 1.0.

## Depression

Information on physician-diagnosed depression was extracted from RAMQ databases (population registry, hospital discharge abstracts and reimbursement claims for consultations with physicians). Hospital admission or consultation dates and ICD-9 codes were obtained from the hospital discharge abstracts and the reimbursement database, respectively. We considered ICD-9 codes 296.x, 300.4, and 311 to define cases (S2 Table). More than 95% of the reimbursement claims made during the years of interest contained an ICD code.

Aligned with previous studies that validated physician-diagnosed depression [26,27], prevalent cases of depression were defined as those with at least one code related to depression during the year prior to participation in $T_2$ and were excluded (see S1 Fig), while participants who had at least one code related to depression after participation in $T_2$ were classified as incident cases. The follow-up period of three years was chosen in agreement with the median follow-up time of three to four years used in previous studies on the effect of psychosocial stressors at work on depression [13,15], We censored all participants who were free of depression at the end of this period.

Deaths during follow-up, obtained from the Demographic Event Registry of the *Institut de la statistique du Québec*, were considered competing risks.

## Covariables

We adjusted the analysis for sex and time-varying potential confounders measured at $T_1$ and $T_2$. These consisted of sociodemographic factors [age (continuous), marital status (married or living together vs. living alone), presence of children in the household], health behaviors [smoking (never, in the past, currently), alcohol consumption ($<$6 drinks/ week, $\geq$ 6 drinks /week), leisure time physical activity ($\leq$ once/month, $\leq$ once/week, $\geq$ once/week)] and pre-existing job strain measured at $T_1$.

## Multiple imputation

In the analytical sample of 6521 participants, there were 5898 complete cases (see S1 Fig). We used multiple imputation using chained equations (MICE) to treat missing information (20 data set, 10 iterations). In each iteration, missing values were first imputed for $T_1$ and then for $T_2$.

## Mediation analysis

We have developed a mediation method for a time-to-event outcome (in our study, onset of depression). The total effect of socioeconomic indicators in $T_1$, on the incidence of depression during the three years following $T_2$, can be decomposed into two parts that are called the natural direct effect and the natural indirect effect. The natural direct effect would be observed if we set psychosocial stressors at work in $T_2$ for each individual to the value they would naturally have if they were not exposed to low SES [28]. The natural indirect effect is the other part of the total effect, i.e. the part due to the change in these stressors caused by low SES.

Older forms of mediation analysis, e.g., the difference of coefficients estimated from nested linear models, depend on strong assumptions to have a causal interpretation. These assumptions are difficult to satisfy in observational studies, especially the assumption that potential confounders of the mediator-outcome relationship are not themselves consequences of the exposure [28].

To obtain causal estimates under less restrictive assumption, we here estimated randomized interventional analogues of the natural direct effect and of the natural indirect effect. Our method extends the approach previously proposed by VanderWeele and Tchetgen Tchetgen [29], to the time-to-event case (see Mathematical Appendix in S1 Appendix); see S2 Fig for a causal diagram of the relations between exposures, mediators, potential time-varying confounders and depression based on the literature. This approach allows controlling for time-dependent confounding, even when a confounder of the mediator-outcome relationship is affected by the exposure. Briefly, the method combines the results of two marginal structural models: one for the outcome, according to both exposure and mediator, and one for the mediator, according to the exposure. Each model seeks to simulate the result of a randomized

experiment (one where the exposure and mediator are randomized, and one where only the exposure is randomized) with respect to the measured covariates, using an inverse probability weighting estimator. Exposure weights were estimated as the inverse of the probability of being exposed to low SES according to a multinomial model adjusted for age, and stratified or adjusted for sex. Similarly, mediator weights were obtained as the inverse of the probability of psychosocial stressors at work in a logistic regression model, adjusted for job strain and SES at $T_1$, and the aforementioned covariates at $T_1$ and $T_2$. The parameters of the marginal structural model for the outcome were estimated using a Cox proportional hazard model including only the mediator and the exposure as independent variables, with observations weighted according to the product of the exposure and mediator weights. Similarly, the marginal structural model for the mediator was estimated using a logistic regression of the mediator according to the exposure, weighted according to the exposure weights. The interventional direct effects (IDE) and interventional indirect effects (IIE) are estimated on the hazard ratio (HR) scale.

Total, direct, and indirect effects were estimated in each imputed dataset along with their standard errors, which were estimated by taking 100 bootstrap samples. The estimates were then pooled across the 20 imputed data sets. In complete case analysis, 2000 bootstrap samples were used. Furthermore, the mediated fraction was estimated using the formula.

$$HR_{IDE} \times [HR_{IIE} - 1]/(HR_{IDE} \times HR_{IIE} - 1)$$

[30].

### Inverse probability of censoring weights

Inverse probability of censoring weights (IPCW) were used to obtain estimates applicable to the initial population at $T_1$ (9188 participants). In this context, censoring was defined as death before $T_2$; impossibility to link with administrative data; attrition or labor force exit before $T_2$; or occurrence of physician-diagnosed depression in the year preceding $T_2$. The weights were obtained through a logistic model, where the probability of censoring was regressed on age, SES indicators and health habits at $T_1$, and stratified or regressed on sex.

### Sensitivity analyses

As a sensitivity analysis, we repeated the mediation analysis excluding all prevalent cases of depression since 1991 instead of excluding only cases that occurred in the year prior to $T_2$. We also performed the mediation analysis with both sexes together when considering as mediators the underlying dimensions psychological job demands, job control, and reward.

All analyses were conducted within R, version 4.1.1, with the *tidyverse* [31], *mlogit* [32], *naniar* [33], *rms* [34], *boot* [35] and *mice* [36] packages.

## Results

S3 Table shows characteristics of the PROQ population at baseline by socioeconomic levels. Important differences were observed between the sexes for all indicators. For example, the proportion of managers was 5 times higher among men than among women, while three quarters of women had non-manager, non-professional occupations. In the analytical sample of 6521 participants, the prevalence of job strain at $T_2$ was 18.0% (women: 20.8%, men: 15.0%) and that of ERI 24.6% (women: 24.0%, men: 25.2%).

Over the following three years, we identified 469 new cases of physician-diagnosed depression, 311 among 3304 women and 158 among 3217 men (33.1 vs. 16.8 per 1000 person-years, respectively). Overall, depression was therefore almost 2 times higher among women than among men, but SES indicators were more strongly related to its incidence in

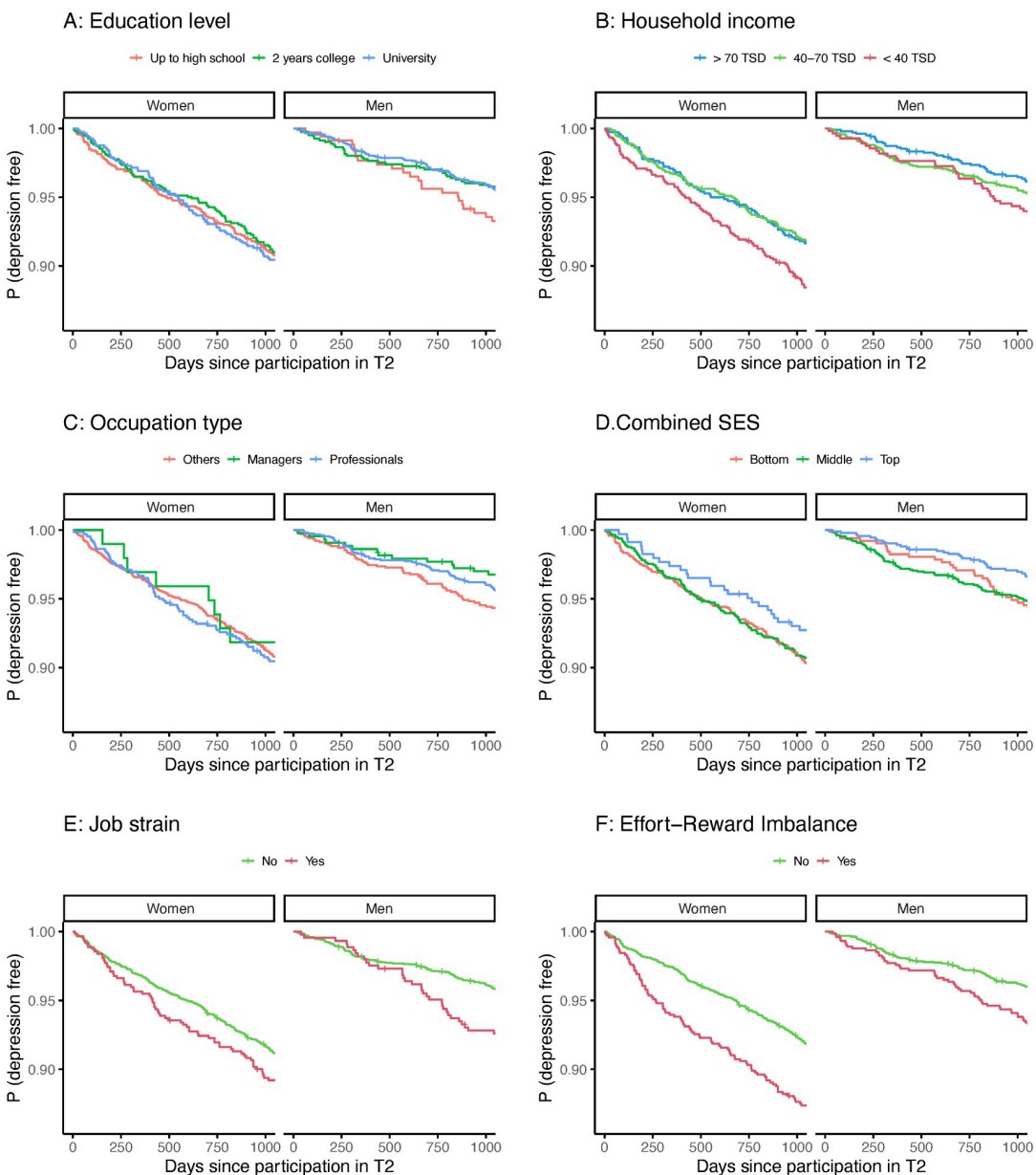

**Fig 1. Kaplan-Meier plots of depression by risk factors.** The population was weighted for age at baseline. TSD: thousands of Canadian dollars.

men (Fig 1A–1D; Table 1 and S4 Table). For example, among men, the HR for depression was 2.1 (95% CI: 1.1–4.2) for non-professional, non-manager occupations, compared with managers, whereas among women this HR was 1.3 (95% CI: 0.6–2.9; Table 1 and S4 Table).

Monotonic trends of increasing risk of depression with decreasing SES were seen in men for all three indicators and for the combined SES (Table 1 and S3 Table). For example, the HR for the middle level of occupational type in men is almost at the midpoint between those for the highest and the lowest level (1.5, 95% CI: 0.8–2.9).

In women, the confidence intervals include the null value and therefore do not permit any conclusions about an effect of the lower levels of SES indicators on the risk of depression.

**Table 1. Total effect estimates of socioeconomic factors and of psychosocial stressors at work on depression in the three years after $T_2$.**

| SES | Men | Women | Both |
|---|---|---|---|
| **Education**[a] | | | |
| Ref: university | 1 | 1 | 1 |
| 2 years college | 1.028 (0.677–1.561) | 0.870 (0.643–1.175) | 0.926 (0.729–1.177) |
| no college | **1.717** (1.080–2.728) | 0.932 (0.702–1.237) | 1.176 (0.912–1.516) |
| **Income**[a] | | | |
| Ref: ≥70 TSD | 1 | 1 | 1 |
| 40–70 TSD | 1.200 (0.819–1.757) | 0.982 (0.722–1.334) | 1.066 (0.835–1.359) |
| < 40 TSD | **1.667** (1.043–2.665) | 1.179 (0.862–1.612) | **1.319** (1.011–1.721) |
| **Occupation**[a] | | | |
| Ref: managers | 1 | 1 | 1 |
| professionals | 1.508 (0.773–2.943) | 1.400 (0.616–3.180) | 1.732 (0.978–3.064) |
| others | **2.129** (1.082–4.188) | 1.318 (0.595–2.918) | **1.886** (1.084–3.280) |
| **Comb. SES**[a] | | | |
| Ref: high | 1 | | 1 |
| medium | 1.469 (0.989–2.183) | 1.222 (0.815–1.831) | 1.303 (0.955–1.779) |
| Low | **1.684** (1.011–2.805) | 1.207 (0.807–1.806) | 1.339 (0.969–1.850) |
| **PSW** | | | |
| **Job strain**[b] | | | |
| Ref: no strain | 1 | 1 | 1 |
| High strain | **1.722** (1.149–2.581) | 1.296 (0.988–1.699) | **1.422** (1.138–1.777) |
| **ERI**[b] | | | |
| Ref: no imbalance | 1 | 1 | 1 |
| Imbalance | **1.569** (1.084–2.273) | **1.834** (1.428–2.356) | **1.729** (1.414–2.115) |

All values are HR. TSD: CAD 1000. Income is before-tax household income per year. **Bold**: 95% CI that do not include 1.

[a] Adjusted for age and (last column only) for sex.

[b] Adjusted for age, (last column only) sex, SES indicators at $T_1$, family indicators (marital status, presence of children in the household) and lifestyle habits (smoking, alcohol consumption and leisure time physical activity) at $T_1$ and $T_2$.

There were also no consistent trends in the relationship between any SES indicator and risk of depression (Fig 1A–1D; Table 1 and S4 Table).

Psychosocial stressors at work were associated with the incidence of depression among men (job strain: HR = 1.7, 95% CI: 1.1–2.6; ERI: HR = 1.6, 95% CI: 1.1–2.3) and in the entire population (Table 1; Fig 1E and 1F). Among women, ERI was strongly associated with the incidence of depression (HR = 1.8, 95% CI: 1.4–2.4), while the evidence for an association with job strain was weaker (Fig 1F; Table 1; S5 Table). When controlling only for age and sex, results were very similar (S5 Table).

We also estimated the association between SES indicators at $T_1$ and PSW at $T_2$ for men (S6 Table), women (S7 Table) and both sexes (S8 Table). Lower SES was consistently associated with higher job strain, as expected, especially in the case of household income and occupation. On the other hand, lower SES was consistently associated with lower ERI.

Finally, Tables 2–4 show estimates of the total and direct effects of SES on depression and of the indirect effects mediated by psychosocial stressors at work, as well as the estimated fraction mediated. For main analyses, the importance of job strain and ERI were investigated separately for men (Table 2), for women (Table 3) and for both sexes together (Table 4).

**Table 2. Total and direct effect estimates of socioeconomic status on depression and indirect effect estimates mediated through psychosocial stressors at work in men.**

| SES | Job strain | | | | ERI | | | |
|---|---|---|---|---|---|---|---|---|
| | Total IE | Direct IE | Indirect IE | % | Total IE | Direct IE | Indirect IE | % |
| **Education** | | | | | | | | |
| Ref: univ. | 1 | | | | 1 | | | |
| 2 yr college | 1.457 (0.797 2.661) | 1.450 (0.797–2.639) | 1.005 (0.977–1.033) | - | 1.221 (0.762–1.957) | 1.262 (0.785–2.029) | 0.968 (0.930–1.006) | - |
| no college | 1.653 (0.793–3.445) | 1.637 (0.786–3.407) | 1.010 (0.974–1.047) | 2.5 | **2.271 (1.282–4.022)** | **2.416 (1.332–4.381)** | 0.940 (0.877–1.007) | - |
| **Income** | | | | | | | | |
| Ref:≥70TSD | 1 | | | | 1 | | | |
| 40–70 TSD | 1.087 (0.774–1.526) | 1.076 (0.765–1.513) | 1.010 (0.997–1.023) | 22 | 1.180 (0.749–1.859) | 1.199 (0.760–1.891) | 0.984 (0.956–1.013) | - |
| < 40 TSD | 1.410 (0.988–2.014) | 1.378 (0.964–1.969) | 1.023 (0.999–1.048) | 11 | **2.090 (1.234–3.542)** | **2.102 (1.241–3.560)** | 0.995 (0.962–1.028) | - |
| **Occupation** | | | | | | | | |
| Ref: manag. | 1 | | | | 1 | | | |
| profession. | 1.992 (0.664–3.774) | 1.937 (0.739–5.076) | 1.028 (0.986–1.073) | 5.6 | 1.680 (0.801–3.521) | 1.713 (0.815–3.598) | 0.981 (0.952–1.010) | - |
| others | **2.837 (1.064–7.565)** | 2.725 (0.747–4.254) | 1.041 (0.984–1.101) | 6.1 | **3.022 (1.437–6.353)** | **3.085 (1.461–6.513)** | 0.980 (0.950–1.010) | - |
| **Comb. SES** | | | | | | | | |
| Ref: high | 1 | | | | 1 | | | |
| median | 1.400 (0.773–2.536) | 1.374 (0.757–2.495) | 1.019 (0.988–1.051) | 6.5 | **1.633 (1.024–2.602)** | **1.662 (1.042–2.650)** | 0.983 (0.957–1.009) | - |
| low | 1.587 (0.769–3.274) | 1.523 (0.741–3.130) | 1.042 (0.993–1.094) | 11 | **2.228 (1.235–4.022)** | **2.303 (1.263–4.200)** | 0.968 (0.927–1.010) | - |

All values are HR, adjusted for age, job strain at $T_1$, family indicators (marital status, presence of children in the household) and lifestyle habits (smoking, alcohol consumption and leisure time physical activity) at $T_1$ and $T_2$. **Bold**: 95% CI that do not include 1. TSD: CAD 1000. Income is before-tax household income per year. IE: Interventional effect.

In all cases, the direct effect estimates were very close to the total effect estimates of SES. Job strain showed a slight, but consistent tendency to mediate an increased incidence of depression. In men, the estimated indirect effect of the lowest levels of SES through job strain were HR = 1.01 (95% CI 0.97–1.05; 3% of total effects) for no college education, HR = 1.02 (95% CI 1.00–1.05; 11% of total effects) for the lowest income level, and HR = 1.04 (95% CI 0.98–1.10; 6% of total effects) for non-manager, non-professional occupation. For the lowest tertile of combined SES, a HR of 1.04 (95% CI 0.99–1.09; 11% total effects) was estimated (Table 2).

The estimates of indirect effects of job strain in women were similar to those in men (Table 3), though results were inconclusive regarding the association between low SES itself and depression (Table 1). We repeated all analyses for the complete cases (5898 participants, see S1 Fig). The overall results led to qualitatively similar conclusions as those obtained in the main analyses (S9–S11 Tables).

The sensitivity analysis excluding all prevalent cases of depression since recruitment produced similar results to those of the main analysis (S12–S14 Tables). In the analysis using the dimensions of the psychological models (S15 Table), reward showed a consistent tendency towards an indirect effect. The estimated indirect effect of the lowest levels of SES were

**Table 3. Total and direct effect estimates of socioeconomic status on depression and indirect effect estimates mediated through psychosocial stressors at work in women.**

| SES | Job strain | | | | ERI | | | |
|---|---|---|---|---|---|---|---|---|
| | Total IE | Direct IE | Indirect IE | % | Total IE | Direct IE | Indirect IE | % |
| **Education** | | | | | | | | |
| Ref: univ. | 1 | | | | 1 | | | |
| 2 yr college | 0.785 (0.538–1.145) | 0.778 (0.532–1.138) | 1.009 (0.993–1.025) | | 0.885 (0.627–1.249) | 0.897 (0.636–1.263) | 0.987 (0.961–1.013) | - |
| no college | 1.032 (0.714–1.487) | 1.025 (0.710–1.480) | 1.005 (0.993–1.017) | 32 | 1.022 (0.736–1.419) | 1.074 (0.773–1.492) | **0.952 (0.920–0.985)** | - |
| **Income** | | | | | | | | |
| Ref:≥70TSD | 1 | | | | 1 | | | |
| 40–70 TSD | 1.027 (0.678–1.556) | 1.017 (0.669–1.546) | 1.010 (0.992–1.029) | 21 | 0.925 (0.628–1.362) | 0.953 (0.649–1.399) | 0.971 (0.941–1.001) | - |
| < 40 TSD | 1.281 (0.842–1.951) | 1.265 (0.829–1.932) | 1.013 (0.991–1.035) | 6.0 | 1.118 (0.773–1.617) | 1.124 (0.778–1.624) | 0.995 (0.966–1.026) | - |
| **Occupation** | | | | | | | | |
| Ref: manag. | 1 | | | | 1 | | | |
| profession. | 1.030 (0.317–3.345) | 1.005 (0.302–3.345) | 1.025 (0.963–1.091) | 22 | 1.274 (0.473–3.426) | 1.341 (0.504–3.567) | 0.950 (0.882–1.022) | - |
| Others | 1.047 (0.336–3.260) | 1.004 (0.309–3.265) | 1.043 (0.962–1.130) | 91 | 1.348 (0.526–3.456) | 1.435 (0.569–3.620) | 0.939 (0.872–1.012) | - |
| **Comb. SES** | | | | | | | | |
| Ref: high | 1 | | | | 1 | | | |
| median | 1.435 (0.808–2.548) | 1.425 (0.797–2.547) | 1.007 (0.984–1.030) | 2.6 | 1.188 (0.751–1.879) | 1.212 (0.770–1.909) | 0.980 (0.944–1.019) | - |
| Low | 1.468 (0.829–2.599) | 1.457 (0.817–2.598) | 1.008 (0.984–1.032) | 2.7 | 1.253 (0.785–2.000) | 1.323 (0.834–2.101) | **0.947 (0.905–0.991)** | - |

All values are HR, adjusted for age, job strain at $T_1$, family indicators (marital status, presence of children in the household) and lifestyle habits (smoking, alcohol consumption and leisure time physical activity) at $T_1$ and $T_2$. **Bold**: 95% CI that do not include 1. TSD: CAD 1000. Income is before-tax household income per year. IE: Interventional effect.

HR = 1.02 (95% CI 1.00–1.03; 10% of total effects) for no college education, HR = 1.04 (95% CI 1.02–1.06; 15% of total effects) for the lowest income level, and HR = 1.06 (95% CI 1.01–1.11; 12% of total effects) for non-manager, non-professional occupation.

## Discussion

The present study examined the adverse effect of low SES and psychosocial stressors at work on the incidence of physician-diagnosed depression. Our results support the hypothesis that low SES, especially in men, and psychosocial stressors at work are both risk factors for depression. Furthermore, this is the first study to decompose the total estimated effects of low SES, using randomized interventional analogues, into direct and indirect effects mediated by psychosocial stressors at work. Our results suggest there may be a modest mediating effect of job strain and low reward, at least in men. We did not find any evidence that the effects of low SES are mediated by ERI.

### Socioeconomic status and depression

We found a strong disparity in the distribution of depression: the lower the level of education, household income, occupation type or combined SES, the higher the incidence of

**Table 4. Total and direct effect estimates of socioeconomic status on depression and indirect effect estimates mediated through psychosocial stressors at work in both men and women.**

| SES | Job strain | | | | ERI | | | |
|---|---|---|---|---|---|---|---|---|
| | Total IE | Direct IE | Indirect IE | % | Total IE | Direct IE | Indirect IE | % |
| **Education** | | | | | | | | |
| Ref: univ. | 1 | | | | 1 | | | |
| 2 yr college | 0.959 (0.685–1.344) | 0.951 (0.679–1.333) | 1.009 (0.993–1.025) | - | 1.030 (0.317–3.345) | 1.005 (0.302–3.345) | 1.025 (0.963–1.091) | |
| no college | 1.228 (0.877–1.718) | 1.219 (0.873–1.702) | 1.007 (0.989–1.025) | 3.7 | **1.451 (1.053–1.972)** | **1.531 (1.105–2.114)** | **0.948 (0.918–0.974)** | - |
| **Income** | | | | | | | | |
| Ref:≥70TSD | 1 | | | | 1 | | | |
| 40–70 TSD | 1.091 (0.771–1.542) | 1.080 (0.763–1.529) | 1.010 (0.997–1.024) | 15 | 1.005 (0.762 1.331) | 1.029 (0.775–1.361) | **0.976 (0.957–0.995)** | - |
| < 40 TSD | 1.411 (0.985–2.022) | 1.379 (0.962–1.976) | 1.024 (1.000–1.048) | 7.9 | **1.369 (1.009–1.856)** | **1.374 (1.014–1.862)** | 0.996 (0.977–1.014) | - |
| **Occupation** | | | | | | | | |
| Ref: manag. | 1 | | | | 1 | | | |
| profession. | 1.583 (0.664–3.774) | 1.546 (0.636–3.758) | 1.024 (0.986–1.063) | 6.3 | 1.729 (0.907–3.298) | 1.782 (0.937–3.390) | 0.970 (0.937–1.000) | - |
| others | 1.847 (0.803–4.249) | 1.573 (0.747–4.254) | 1.036 (0.975–1.100) | 7.5 | **2.228 (1.199–4.140)** | **2.305 (1.244–4.270)** | **0.968 (0.936–0.998)** | - |
| **Comb. SES** | | | | | | | | |
| Ref: high | 1 | | | | 1 | | | |
| median | 1.437 (0.922–2.240) | 1.418 (0.905–2.223) | 1.013 (0.993–1.033) | 4.3 | 1.347 (0.949–1.913) | 1.373 (0.968–1.948) | 0.981 (0.962–1.001) | - |
| low | 1.517 (0.958–2.402) | 1.490 (0.933–2.378) | 1.018 (0.990–1.047) | 5.4 | 1.468 (1.022–2.159) | 1.551 (1.066–2.256) | 0.958 (0.931–0.985) | - |

All values are HR, adjusted for age, sex, job strain at $T_1$, family indicators (marital status, presence of children in the household) and lifestyle habits (smoking, alcohol consumption and leisure time physical activity) at $T_1$ and $T_2$. **Bold**: 95% CI that do not include 1. TSD: CAD 1000. Income is before-tax household income per year. IE: Interventional effect.

depression, mainly among men. In line with these findings, systematic reviews found consistent evidence for associations between socioeconomic disadvantage and depression [5,6,37,38].

When we compared the incidence of depression by sex, women had higher rates than men at all levels of the SES indicators (Fig 1A–1D). At the same time, among women, the estimated benefits of a high SES on depression were consistently weaker than those estimated among men. We even found a tendency towards higher rates of depression for women with university degrees compared to those with less education (Table 1). This might be explained if high SES women suffered a particular pressure to perform as well as or better than their male colleagues [39], possibly in combination with higher expectations for family and domestic work. In the present study, the fraction of women at the highest levels of education and occupation type was rather low (26.5% with university education, 3.5% managers, see S3 Table).

Few studies on the topic have analyzed women and men separately, and findings were inconsistent. In a recent systematic review, only three among 23 studies compared the socioeconomic gradient of depression between women and men in adults [5]. One of them found a stronger gradient in men [40], one in women [41], and one found no difference [42].

## Socioeconomic status and psychosocial stressors at work

In the present study, low SES was associated with a higher prevalence of job strain and a lower prevalence of ERI (S5–S7 Tables). The association between low SES and increased job strain is consistent with previous evidence [9,43]. On the other hand, the direction of association between low SES and ERI is inconsistent in the literature [9,44,45] with some studies reporting associations similar to those shown here [43,46]. This variability could be partially explained by the fact that, while low SES workers have been shown to be more frequently exposed to low rewards, especially its subcomponent job insecurity [44,45,47], this is, in some cohorts, counterbalanced by their lower exposure to perceived efforts [44,47]. Therefore, results for a mediating effect using low reward, but not ERI, could be explained by its specific association with low SES.

## Psychosocial stressors at work and depression

Our results showed an association of both job strain and of ERI with the incidence of physician-diagnosed depression in the total population. The present estimate for the effect of job strain [HR = 1.4 (95% CI 1.1–1.8)] is somewhat lower than the pooled value of RR = 1.8 (95% CI 1.5–2.1) obtained in a recent meta-analysis [13]. It is noteworthy that the present study excluded cases of depression that appeared before measurement of psychosocial stressors at work, reducing the probability of reverse causation.

The estimated effect of ERI found here [HR = 1.7 (95% CI 1.4–2.1)] is somewhat higher than the pooled estimate obtained in the most recent meta-analysis [RR = 1.5 (95% CI 1.3–1.8)] [15]. Previous meta-analyses [13,15,16] included studies where depression was defined mainly from self-report in questionnaires.

No consistent conclusions on the relation between psychosocial stressors at work and depression in sex strata can be drawn from the literature [13]. In the present study, we found that, among men, both job strain and ERI were associated with the incidence of depression, while among women the evidence for an association was stronger in the case of the ERI model. With regard to job strain, Shields et al. found similar results to ours in their most adjusted analysis: clear evidence for an association between job strain and depression in men [OR = 1.7 (95% CI 1.2–2.5)], but not in women [OR = 1.1 (95% CI 0.7–1.5)] [48].

## Mediation

Regarding causal mechanisms for the socioeconomic gradient of depression, it appears that the effects of low levels of education, income and occupation on the incidence of depression may primarily result from pathways that do not pass through stressors at work. Job strain and low reward may mediate only a modest fraction of the total effects. Moreover, the estimated indirect effects of low SES mediated by ERI tended to reduce the total effects.

We have found only two previous prospective studies that have examined this question. Re-analysis of results published by Theorell et al. (2012) suggests a possible mediating role of psychological job demands and job control, although it is not possible to derive confidence intervals for this indirect effect [18]. Confidence intervals for a possible mediating effect of job strain in Hoven et al. (2015) included the null value, while that study did find evidence that ERI mediated part of the estimated effect of low occupational status [17]. It may be relevant to point out that the present study avoided a possible reverse causation bias by measuring the mediators later than the exposition, while the previous studies measured both at the same point in time.

## Strengths and limitations

Among the strengths of the present study are the prospective study design and the use of validated instruments and algorithms for measuring job strain and depression. Physician-diagnosed depression is not subject to confirmation bias in questionnaires, to recall bias, or to differential misclassification [26]. Most studies on inequality use a single indicator of SES, but each indicator provides only partial insight into the structures of social hierarchies; for example, legal immigrants are on average more highly educated, but less well remunerated and occupy less prestigious positions than native-born citizens [49]. To increase insight, we have shown results for the three main indicators of SES, separately and combined. The study design respects the temporal order of exposure, mediator and outcome, and we have used appropriate statistical methods to deal with the uncertainty deriving from missing data and attrition. Furthermore, we have chosen a method that is robust against the effect of the exposure on a confounder of the mediator-outcome relation and adapted it here to a time-to-event outcome.

Some limitations need to be considered when appraising these results. For analyses involving ERI, we could not control for previous exposure at baseline (1991–1993), since this model was first published in 1996 [8]. Therefore, we cannot exclude the possibility that some of the cases of ERI ascertained at $T_2$ might have already been present at baseline and might not be due to low SES. Such an error would be expected to overestimate the indirect effect. As next best alternative, we have adjusted for job strain at baseline. On the other hand, our use of a proxy scale for effort in measuring ERI at $T_2$ might introduce a non-differential classification error, which we expect to result in an underestimation of the indirect effect.

In general, there may be unknown confounders, such as genetic or familial predispositions towards depression or personality types, which, given the observational design, might be associated with the exposure, the mediators, and the outcome. In addition, healthy worker bias is inherent to working population cohorts [50]: potential participants with stronger socioeconomic disadvantages or higher levels of psychosocial stressors at work may have been censored at baseline.

Regarding the outcome, the estimation of depression from administrative data may underestimate true incidence, since it depends on factors such as case detection by the medical professional, emotional self-knowledge and understanding on the part of the patient, active search for professional help, and access to medical support and treatment. Specifically, it might be suspected that the higher incidence observed in women might be due to more help-seeking and more awareness of depression than in men [51]. However, in the same cohort, we have previously shown that the concordance between cases of depression defined from administrative and from questionnaire data does not vary between men and women [26], militating against such an explanation. On the other hand, misclassification might have arisen from our previously validated definition of cases, which included administrative codes for episodes of manic/bipolar disorders [26]. These cases, however, represented only 3.6% of the total number of cases (S2 Table). Their inclusion is therefore not expected to modify our conclusions.

Help-seeking behavior and recognition of depression may be negatively associated with low SES [51]. However, Quebec has a universal health care system that reduces disparities in access, and we and others have shown previously that the concordance between administrative and questionnaire data does not vary across SES strata nor between areas of lower and higher SES within Quebec [26,52].

Finally, caution should be taken when generalizing the present results, as the study population was composed almost entirely of white-collar workers in stable employment, even though we found a considerable degree of diversity in their SES indicators. The cohort was constituted in 1991 and follow-up ended in 2006. Future studies on depression may deal with labor

relationships that have become more unstable in the intervening time, while socioeconomic inequality have become more pronounced [53]. This process has been catalyzed in the last years by the Covid-19 pandemic [54], suggesting that socioeconomic gradients and the effects of psychosocial stressors at work might become even more important.

## Conclusion

Our results show that socioeconomic inequalities in working life, especially among men, and psychosocial stressors at work increase the incidence of depression and that the latter only slightly mediate the socioeconomic gradient of depression. Inequalities and psychosocial work conditions both require interventions if one desires to promote mental health and reduce the incidence of depression.

## Supporting information

**S1 Fig. Flowchart of the PROspective Quebec (PROQ) Study on Work and Health for the current mediation analyses.** [1] Participants missing the necessary information to be re-contacted at follow-up. [2] 6521/7528 = 86.6% of eligible at $T_2$; 6521/9188 = 71.0% of baseline; men: 3217, women: 3304 (50.7%). [3] Not mutually exclusive, total = 623 participants. [4] 5898/7528 = 78.3% of eligible at $T_2$; 5898/9188 = 64.2% of baseline; men: 2963, women: 2935 (49.8%).
(TIF)

**S2 Fig. Directed acyclic graph for the mediation analyses.** *SES*: Socioeconomic status at $T_1$. *Cov*: Covariates (potential confounders) at $T_1$ and $T_2$. *PSW*: Psychosocial stressors at work at $T_2$. *Dep*: Depression registered over the 3 years following $T_2$.
(TIF)

**S1 Table. Combined measure of socioeconomic status using principal component analysis (n = 5898 complete cases).** [a] Three levels as in main analyses (1: < 40 000 CAD, 2: 40 000–70 000 CAD, 3: ≥ 70 000 CAD). [b] Three levels as in main analyses (1: Without college, 2: 2 years college, 3: Bachelor). [c] Three levels as in main analyses (1: Others, 2: professionals, 3: Managers). For all further analyses, component 1 was used as combined measure of socioeconomic status.
(PDF)

**S2 Table. ICD-9 codes for identifying cases of physician-diagnosed depression.**
(PDF)

**S3 Table. S3a Table**: Cohort characteristics at baseline (1991–1993) by educational level. **S3b Table**: Cohort characteristics at baseline (1991–1993) by household income. **S3c Table**: Cohort characteristics at baseline (1991–1993) by occupation type.
(PDF)

**S4 Table. Socioeconomic gradient of depression in the three years after $T_2$ (n = 5898 complete cases).** $n_{dep}$: Number of cases of depression. PY: Person-years. All other values are HR, adjusted for age and for sex (last column). **Bold**: 95% CI that do not include 1. TSD: 1000 CAD$. Income is before tax household income per year.
(PDF)

**S5 Table. Effect estimates of psychosocial stressors at work on depression in the three years after $T_2$ (n = 5898 complete cases).** $n_{dep}$: Number of cases of depression. PY: Person-years. All other values are HR. Model 1: Adjusted for age and for sex. Model 2: Adjusted for

age, sex (last column), SES indicators at $T_1$, family indicators (marital status, presence of children in the household) and lifestyle habits (smoking, alcohol consumption and leisure time physical activity) at $T_1$ and $T_2$. **Bold**: 95% CI that do not include 1.
(PDF)

**S6 Table. Association between SES and psychosocial stressors at work in men (n = 2963 complete cases).** All values are HR, adjusted for age. **Bold**: 95% CI that do not include 1. TSD: CAD 1000. Income is before-tax household income per year.
(PDF)

**S7 Table. Association between SES and psychosocial stressors at work in women (n = 2935 complete cases).** All values are HR, adjusted for age. **Bold**: 95% CI that do not include 1. TSD: CAD 1000. Income is before-tax household income per year.
(PDF)

**S8 Table. Association between SES and psychosocial stressors at work in both men and women (n = 5898 complete cases).** All values are HR, adjusted for age and sex. **Bold**: 95% CI that do not include 1. TSD: CAD 1000. Income is before-tax household income per year.
(PDF)

**S9 Table. Contribution of psychosocial stressors at work to the socioeconomic gradient of depression in men (n = 2963 complete cases).** All values are HR, adjusted for age, job strain at $T_1$, family indicators (marital status, presence of children in the household) and lifestyle habits (smoking, alcohol consumption and leisure time physical activity) at $T_1$ and $T_2$. **Bold**: 95% CI that do not include 1. TSD: CAD 1000. Income is before-tax household income per year. IE: Interventional effect.
(PDF)

**S10 Table. Contribution of psychosocial stressors at work to the socioeconomic gradient of depression in women (n = 2935 complete cases).** All values are HR, adjusted for age, job strain at $T_1$, family indicators (marital status, presence of children in the household) and lifestyle habits (smoking, alcohol consumption and leisure time physical activity) at $T_1$ and $T_2$. **Bold**: 95% CI that do not include 1. TSD: CAD 1000. Income is before-tax household income per year. IE: Interventional effect.
(PDF)

**S11 Table. Contribution of psychosocial stressors at work to the socioeconomic gradient of depression in both men and women (n = 5898 complete cases).** All values are HR, adjusted for age, sex, job strain at $T_1$, family indicators (marital status, presence of children in the household) and lifestyle habits (smoking, alcohol consumption and leisure time physical activity) at $T_1$ and $T_2$. **Bold**: 95% CI that do not include 1. TSD: CAD 1000. Income is before-tax household income per year. IE: Interventional effect.
(PDF)

**S12 Table. Total and direct effect estimates of socioeconomic status on depression and indirect effect estimates mediated through psychosocial stressors at work in men after excluding prevalent cases since 1991.** All values are HR, adjusted for age, job strain at $T_1$, family indicators (marital status, presence of children in the household) and lifestyle habits (smoking, alcohol consumption and leisure time physical activity) at $T_1$ and $T_2$. **Bold**: 95% CI that do not include 1. TSD: CAD 1000. Income is before-tax household income per year. IE: Interventional effect.
(PDF)

**S13 Table. Total and direct effect estimates of socioeconomic status on depression and indirect effect estimates mediated through psychosocial stressors at work in women after excluding prevalent cases since 1991.** All values are HR, adjusted for age, job strain at $T_1$, family indicators (marital status, presence of children in the household) and lifestyle habits (smoking, alcohol consumption and leisure time physical activity) at $T_1$ and $T_2$. **Bold**: 95% CI that do not include 1. TSD: CAD 1000. Income is before-tax household income per year. IE: Interventional effect.
(PDF)

**S14 Table. Total and direct effect estimates of socioeconomic status on depression and indirect effect estimates mediated through psychosocial stressors at work in both men and women after excluding prevalent cases since 1991.** All values are HR, adjusted for age, sex, job strain at $T_1$, family indicators (marital status, presence of children in the household) and lifestyle habits (smoking, alcohol consumption and leisure time physical activity) at $T_1$ and $T_2$. **Bold**: 95% CI that do not include 1. TSD: CAD 1000. Income is before-tax household income per year. IE: Interventional effect.
(PDF)

**S15 Table. Total and direct effect estimates of socioeconomic status on depression and indirect effect estimates mediated through dimensions of psychosocial stressors at work in both men and women.** All values are HR, adjusted for age, sex, job strain at $T_1$, family indicators (marital status, presence of children in the household) and lifestyle habits (smoking, alcohol consumption and leisure time physical activity) at $T_1$ and $T_2$. **Bold**: 95% CI that do not include 1. TSD: CAD 1000. Income is before-tax household income per year. IE: Interventional effect.
(PDF)

**S1 Appendix.**
(PDF)

## Author Contributions

**Conceptualization:** Ana Paula Bruno Pena-Gralle, Denis Talbot, Xavier Trudel, Alain Milot, Mahée Gilbert-Ouimet, Mathilde Lavigne-Robichaud, Ruth Ndjaboué, Alain Lesage, Sophie Lauzier, Michel Vézina, Chantal Brisson.

**Data curation:** Ana Paula Bruno Pena-Gralle, Chantal Brisson.

**Formal analysis:** Ana Paula Bruno Pena-Gralle, Denis Talbot, Johannes Siegrist, Chantal Brisson.

**Funding acquisition:** Denis Talbot, Xavier Trudel, Alain Milot, Mahée Gilbert-Ouimet, Ruth Ndjaboué, Alain Lesage, Sophie Lauzier, Michel Vézina, Chantal Brisson.

**Investigation:** Ana Paula Bruno Pena-Gralle, Denis Talbot, Chantal Brisson.

**Methodology:** Ana Paula Bruno Pena-Gralle, Denis Talbot, Johannes Siegrist.

**Project administration:** Ana Paula Bruno Pena-Gralle, Chantal Brisson.

**Resources:** Ana Paula Bruno Pena-Gralle, Chantal Brisson.

**Software:** Ana Paula Bruno Pena-Gralle, Denis Talbot.

**Supervision:** Xavier Trudel, Alain Milot, Mahée Gilbert-Ouimet, Chantal Brisson.

**Validation:** Ana Paula Bruno Pena-Gralle, Denis Talbot, Mathilde Lavigne-Robichaud, Chantal Brisson.

**Visualization:** Ana Paula Bruno Pena-Gralle, Mathilde Lavigne-Robichaud.

**Writing – original draft:** Ana Paula Bruno Pena-Gralle.

**Writing – review & editing:** Ana Paula Bruno Pena-Gralle, Denis Talbot, Xavier Trudel, Alain Milot, Mahée Gilbert-Ouimet, Mathilde Lavigne-Robichaud, Ruth Ndjaboué, Alain Lesage, Sophie Lauzier, Michel Vézina, Johannes Siegrist, Chantal Brisson.

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
