## [Decision Letter · Decision Letter 0]

6 Jun 2023

PONE-D-23-03329Socioeconomic inequalities, psychosocial stressors at work and physician-diagnosed depression: time-to-event mediation analysis in the presence of time-varying confounders

PLOS ONE

Dear Dr. Pena-Gralle,

Thank you for submitting your manuscript to PLOS ONE. After careful consideration, we feel that it has merit but does not fully meet PLOS ONE’s publication criteria as it currently stands. Therefore, we invite you to submit a revised version of the manuscript that addresses the points raised during the review process.

Unfortunately, despite more than 20 requests, we have received only comments from one reviewer since your submission. Therefore, I have also written a review myself. I would encourage you to revise your manuscript and to address the detailed reviewer comments.

We look forward to receiving your revised manuscript.

Kind regards,

Swaantje Wiarda Casjens

Academic Editor

PLOS ONE

Journal Requirements:

“This work was supported by the Canadian Institutes of Health Research, https://cihr-irsc.gc.ca/e/193.html, Canada [Grant MOP-133542] to CB and by VITAM (Sustainable Health Research Center, https://vitam.ulaval.ca/), Canada to APBPG. DT was supported by a career award from the Fonds de Recherche du Québec – Santé (https://frq.gouv.qc.ca/sante/). The funders had no role in study design, data collection and analysis, decision to publish, or preparation of the manuscript.”

Reviewers' comments:

Reviewer's Responses to Questions

**Comments to the Author**

1. Is the manuscript technically sound, and do the data support the conclusions?

Reviewer #1: Yes

Reviewer #2: Yes

2. Has the statistical analysis been performed appropriately and rigorously? 

Reviewer #1: Yes

Reviewer #2: Yes

3. Have the authors made all data underlying the findings in their manuscript fully available?

Reviewer #1: No

Reviewer #2: No

4. Is the manuscript presented in an intelligible fashion and written in standard English?

Reviewer #1: Yes

Reviewer #2: Yes

5. Review Comments to the Author

Reviewer #1: This is an interesting paper on an important topic; whether the association between low socioeconomic status (SES) and increased risk of depressive disorder is partly mediated by adverse psychosocial working conditions (job stressors). The topic is not new, however, a clear answer has not emerged yet in the literature. The results of this paper suggest that if there is such a mediation, it is only modest. The study has several methodological strengths (temporal order of exposure, mediator, outcome; SES measured with three different indicators, analyzed separately and combined; two job stressors based on major theoretical models, register-based measurement of depressive disorder).

The paper is relatively well-written and the analyses seem to be well-conducted. The authors made several decision when analyzing their data. I want to challenge some of these decisions and I would like to suggest some changes in the analyses.

1) The outcome of the study was physician-diagnosed depressive disorders. You write that you considered ICD-9 codes 296.x, 300.4 and 311.x and ICD-10 codes F30-F34 and F39. With the ICD-9 codes it is complicated, because they use a terminology that is no longer used (such as neurotic depression). For ICD-10 however, there is a clear distinction between Manic episode (F30) and Bipolar affective disorder (F31) on the one hand side and Depressive episode (F32), Recurrent Depressive Disorder (F33) on the other hand. F34 includes both Cyclothymia (34.0) and Dysthymia (34.1). Because you include all these diagnosis, you cannot say that your outcome was depressive disorder. Instead your outcome is Mood disorder/Affective disorder. I suggest that you keep your focus on depressive disorders and that you only include ICD-10 F32, F33 and F34.1.

2) To make this a prospective study, you excluded individuals with a physician-diagnosed depressive disorder in the 12 months before t2. Why only in the 12 months before t2? Why not excluding everyone with a life-time diagnosis of depressive disorder before t2, regardless if this was 1, 5, 10 or 20 years ago? Or, if this is not possible, can you at least exclude individuals up to 12 months before t1? In recent analyses from our group, it made a difference whether we excluded individuals with prevalent depressive disorders during the last year, last 2 years or last 5 years.

3) The causal chain that you want to test is: low SES -> high exposure to stressors at work (job strain, ERI and their components) -> increased risk of depressive disorders. To analyze this you need to measure a) SES at t1, b) job strain and ERI at both t1 and t2 (so you can analyze if there is a prospective association between low SES at t1 and job strain and ERI at t2) and c) the outcome, depressive disorder, from t1 to t3 (end of follow-up) in registers, so you can analyze if there is a prospective association between low SES at t1 and new onset of depressive disorder until t3 and if this association is mediated by job strain/ERI at t2. As delineated above, you measured the outcome only from t2 to t3, and I hope you will change this, to include the outcome measure also at t1 (and ideally even before t1, i.e. life-time history of depressive disorder, so you can exclude everyone with a life-time diagnosis of depressive disorder before t2). However, you do not have data on ERI at t1, you only have data on ERI at t2. For job strain it is different, here you have data on job strain both at t1 and t2 and you adjust your mediation analyses for job strain at t1. But you cannot do this for ERI. This is an important limitation of your mediation analysis that should be addressed in the discussion. You address in the discussion section that you could not adjust for ERI at t1 (page 16, lines 330 to 335) but you are not making clear enough that this is a problem for your mediation analysis. Please make this more clear.

4) Regarding the adjustment for job strain at t1: You state in the method section (page 7, line 144 to 145) that you have adjusted your analyses for “pre-existing job strain measured at t1”. However, in the legends of the tables (table 1 to table 4 and the tables in the supplementary appendix), job strain at t1 is not listed as a covariate. I assume that this is just an omission in the legends, but please check if you have adjusted for job strain at t1 or not in your analyses.

5) For table 1 (and the corresponding supplementary tables), though, that is not about mediation but is about the associations between SES (at t1) and job strain and ERI (at t2) and onset of depressive disorder (after t2), analyses should NOT be adjusted for job strain at t1. This is because you want to show in table 2 the association between job strain/ERI at t2 and risk of depressive disorder after t2. If you adjust for job strain at t1, then you are showing estimates for the change of job strain from t1 to t2 and risk of depressive disorder after t2 and this is, obviously, a different topic. So, please do not adjust the estimates in table 1 for job strain at t1.

6) However, the estimates for job strain and ERI in table 1 should be adjusted for SES. Because SES can easily be a confounder for the association between job strain/ERI and risk of depressive disorder. If you do not want to adjust the estimates for job strain/ERI for SES in table 1, then please do this in a table in the supplementary analysis, but please make sure that there is a table somewhere in the manuscript that shows the estimates for job strain/ERI adjusted for at least sex, age and SES (and other sociodemographic variables and health behaviours, if you want). This will also allow other researchers to include your estimates in future meta-analyses.

7) Table 1: The estimates for the SES measures are adjusted for age and sex, whereas the estimates for job strain/ERI are adjusted for age, sex, marital status, presence of children in the household, smoking, alcohol consumption and physical activity). What is the rational for adjusting SES and job strain/ERI for different sets of covariates?

8) All tables: In the legends you write: “and for lifestyle habits at t1 and t2 (marital status, presence of children in the household, smoking, alcohol consumption and physical activity)”. Sounds a little bit strange to call “marital status” and “presence of children in the household” a “lifestyle habit”. Please consider to revise. Please also clarify if “physical activity” means “leisure time physical activity”.

9) Table 1: Please provide the following information in the table: Number of participants in each exposure group, number of cases in each exposure group, number of cases per 1000 person-years in each exposure group.

10) I am missing a table showing the association between the exposure and the mediator. For example correlations between SES at t1 and job strain at t1, SES at t1 and job strain and ERI at t2, SES at t2 and job strain and ERI at t2. I am also missing an analysis showing whether SES at t1 predicts job strain at t2 (after adjustment for job strain at t1). And maybe also an analysis whether SES at t1 predicts ERI at t2 (after adjustment for job strain at t1). Thus, some data showing that the assumption that exposure to low SES is associated with an increased risk of job stress is confirmed in this study.

Minor comments

11) Introduction, page 3, lines 58-60: “These stressors, in turn, are associated with the onset of several physical and mental health problem (note: a plural “s” is missing here) (references 10-12), including depression (references 13,14). Reference #24 (Rugulies et al 2017) that appear later in manuscript, could be added here to references #13 and #14. Reference #11 (Mikkelsen et al) would fit better with references #13 and #14 than with references #10 and #12. Consider to add to references #10 and #12, a reference to a recent meta-review on job stressors and various health outcomes by Niedhammer et al 2021 (https://www.ncbi.nlm.nih.gov/pubmed/34042163)

12) Methods, page 6, lines 116-118: “Imbalance was considered present when the sum of scores for psychological job demands, a proxy of the Siegrist effort scale (reference 21) was higher than that for reward.” Is this equal to calculating a ratio between efforts (or in this case job demands) and rewards and then using a ratio >1.0 as an indication of ERI? If this is equal then please mention this, because this would mean that you have calculated ERI in a similar way as in several previous studies.

13) Results page 10, line 221-222: “but SES indicators were more strongly related to its incidence in men (Fig. 1A-C)”. Should this be “Fig. 1A-D” instead?

14) Results page 11, line 232: “investigated separately for men (Table 2), for women (Table 3) and for both together (Table 4)”. I suggest to write “for both sexes together (Table 4)”.

15) Results page 11, at several places (and maybe also at other places in the paper): There is the abbreviation “SSE”. What is this? Or is this a typo and should mean SES?

16) Discussion, page 15, lines 303-307: “Regarding causal mechanisms…”This is a very long sentence that is not easy to understand. Please break down into two sentences.

17) Discussion, page 18, line 370-373: “For example, the 2021 Noble prize in economy…”. Please add a reference to the study that demonstrated that increasing the minimum wage had no negative effect on employment.

18) Discussion, page 18, line 373-374: “Regarding psychosocial stressors at work, countries such as Iceland have decreased weekly work hours”. First, if you want to keep this sentence, then please add a reference. Second, I am not sure about the relevance here. In your paper you did not define job stressors by lengths of working hours but by other parameters, so I am wondering about the relevance of this sentence. I suggest that you consider to delete this sentence about working hours in Iceland.

Reviewer #2: The authors used a large Canadian data set to examine the effects of socioeconomic status and psychosocial stressors at work on depression incidence and confirmed the hypothesis that low SES and psychosocial stressors at work are risk factors for depression. Of interest are the results of the mediator analysis, which show a modest mediating effect of job strain and low reward, but no mediating effect by ERI.

The analyses seem to have been well conducted. However, I would like to make a few points:

Methods:

1) Line 99: On what basis were the household income cutoffs chosen?

2) Lines 108-114: Please give more information on the used instruments, i.e. range of values, median of the representative sample of Quebec workers.

3) Lines 115-118: Was the effort part of the ERI not assessed? Please explain. If you have data on effort, please also provide these results and the known ERI.

4) Lines 128-132: It is not clear to me what happened to individuals who received a depression diagnosis more than a year before T2.

5) Line 132: I assume that the second wave and the following wave are different names for the same thing. Is that correct? If yes, a consistent designation would be helpful. If no, please make the difference clearer.

6) Lines 153ff: I miss an introduction to statistical analysis. It should be mentioned that a time series analysis is performed with the dependent variable "depression yes/no". Is this correct? Readers with no prior experience in mediation analysis should be better considered.

7) Lines 155+156: What is a natural direct effect or natural indirect effect? I don't understand the "natural".

Results:

8) How relevant is the analysis of occupation type? Occupation type is very broad and describes very heterogeneous groups. What conclusions can be drawn from such a variable? Since these results are neither described nor discussed, I wonder if these analyses are relevant and should be presented in the main body of the paper.

9) Tables S2: A total column per sex would be convenient in Table S2. A subdivision of S2 into S2a (stratification by education), S2b (stratification by household income), etc. could improve clarity.

10) Line 207: I cannot find ERI in Table S2.

11) Figure 1: Please check if in figure 1B (household income) the graphs for the sexes are not reversed. Because in the tables you see the negative impact of low household income for men and not for women.

12) Lines 219-220: I struggle with your sentence „In women, results were inconclusive: the confidence intervals include both a considerable 220 protection and an increased risk of depression at the lower levels of SES indicators.“.

Isn't it more likely that SES may not have a statistically significant impact on a depression diagnosis in women?

13) Conlcusion: In my opinion, the comments on the 2021 Nobel Prize and iceland are not conclusions. Here, only conclusions of the study should be mentioned and no further discussion should be led.

Minor remarks:

Please remember, abbreviations need to be introduced only once and should be used after their introduction (e.g. ERI or SES).

Line 88: Please explain the abbreviation CHU.

Figure 1: Typo: It should be „sex“

Typo: SSE in several lines

Line 143: Please check your information on alcohol consumption. In Table S2, for example, 6 doses/week was used as the cut-off and here 5 drinks are mentioned.

Line 276: Please provide the percentage of women with the highest level of education and do not just refer to Table S2.

Line 300: Typo 95%IC

6. PLOS authors have the option to publish the peer review history of their article (what does this mean?). If published, this will include your full peer review and any attached files.

Reviewer #1: No

Reviewer #2: No

---

## [Author Response · Author response to Decision Letter 0]

2 Aug 2023

1) The outcome of the study was physician-diagnosed depressive disorders. You write that you considered ICD-9 codes 296.x, 300.4 and 311.x and ICD-10 codes F30-F34 and F39. With the ICD-9 codes it is complicated, because they use a terminology that is no longer used (such as neurotic depression). For ICD-10 however, there is a clear distinction between Manic episode (F30) and Bipolar affective disorder (F31) on the one hand side and Depressive episode (F32), Recurrent Depressive Disorder (F33) on the other hand. F34 includes both Cyclothymia (34.0) and Dysthymia (34.1). Because you include all these diagnosis, you cannot say that your outcome was depressive disorder. Instead your outcome is Mood disorder/Affective disorder. I suggest that you keep your focus on depressive disorders and that you only include ICD-10 F32, F33 and F34.1.

Thank you for your observation. We appreciate the opportunity to provide clarification on our study's methods and limitations. Our complete administrative database includes both ICD-9 and ICD-10 codes. However, all outpatient reimbursement requests in our administrative data (1991-2018) were based on ICD-9 codes. On the other hand, for hospitalized patients, ICD-10 was implemented starting in 2006.

In the current study, we focused on a follow-up period of three years after the second wave of data collection (1999-2001). Consequently, in the present manuscript, no cases were diagnosed using ICD-10, and we have removed all references to ICD-10.

We are aware of the limitations of ICD-9 codes in delimiting depressive disorder from other affective disorders and lament this limitation of the administrative data we used.

2) To make this a prospective study, you excluded individuals with a physician-diagnosed depressive disorder in the 12 months before t2. Why only in the 12 months before t2? Why not excluding everyone with a life-time diagnosis of depressive disorder before t2, regardless if this was 1, 5, 10 or 20 years ago? Or, if this is not possible, can you at least exclude individuals up to 12 months before t1? In recent analyses from our group, it made a difference whether we excluded individuals with prevalent depressive disorders during the last year, last 2 years or last 5 years.

Thank you for bringing these points to our attention. We agree that the ideal design would be to use administrative indicators of depression for the whole life of the subjects, including the time span before T1, but we do not have access to those data. The decision to exclude subjects with physician-diagnosed depression during the 12 months preceding T2 was based on an extensive literature review (see attached document previous_depression.xlsx) and on our own validation study (1). 

However, we recognize that not excluding individuals who had been diagnosed with depression before that 12-month period might introduce some degree of residual confounding. We have therefore now performed a sensitivity analysis where we have excluded all prevalent cases visible in our data, i.e. all cases of depression diagnosed between 1991-01-01 and T2 (new S9-S11 Tables, discussed on p. 10, ll. 207-209 and p. 16, ll. 296-297). The results are not greatly different from the analyses with a 12-month exclusion period and therefore reinforce the overall validity of the conclusions. 

3) The causal chain that you want to test is: low SES -> high exposure to stressors at work (job strain, ERI and their components) -> increased risk of depressive disorders. To analyze this you need to measure a) SES at t1, b) job strain and ERI at both t1 and t2 (so you can analyze if there is a prospective association between low SES at t1 and job strain and ERI at t2) and c) the outcome, depressive disorder, from t1 to t3 (end of follow-up) in registers, so you can analyze if there is a prospective association between low SES at t1 and new onset of depressive disorder until t3 and if this association is mediated by job strain/ERI at t2. As delineated above, you measured the outcome only from t2 to t3, and I hope you will change this, to include the outcome measure also at t1 (and ideally even before t1, i.e. life-time history of depressive disorder, so you can exclude everyone with a life-time diagnosis of depressive disorder before t2). However, you do not have data on ERI at t1, you only have data on ERI at t2. For job strain it is different, here you have data on job strain both at t1 and t2 and you adjust your mediation analyses for job strain at t1. But you cannot do this for ERI. This is an important limitation of your mediation analysis that should be addressed in the discussion. You address in the discussion section that you could not adjust for ERI at t1 (page 16, lines 330 to 335) but you are not making clear enough that this is a problem for your mediation analysis. Please make this more clear.

We agree with the revisor that the ideal design would be to use indicators of depression for the whole life of the subjects including the time span before T1, but we do not have access to those data. We do have administrative indicators of depression for the time span from 1991 to T2 and have now included an additional analysis excluding such prevalent cases (new S9-S11 Tables, discussed on p. 10, ll. 207-208 and p. 16, ll. 296-297; see also our response to comment 2).

We also agree that the impossibility to measure ERI at T1 is a problem for the causal interpretation of any indirect effect through ERI and have made this clearer in the MS (p. 20, ll. 382-385). 

4) Regarding the adjustment for job strain at t1: You state in the method section (page 7, line 144 to 145) that you have adjusted your analyses for “pre-existing job strain measured at t1”. However, in the legends of the tables (table 1 to table 4 and the tables in the supplementary appendix), job strain at t1 is not listed as a covariate. I assume that this is just an omission in the legends, but please check if you have adjusted for job strain at t1 or not in your analyses.

Thank you for noting this oversight. The analyses shown in all tables except Table 1 and Suppl. Tables 3-4 were indeed adjusted for job strain at T1, and we have now corrected the table legends.

5) For table 1 (and the corresponding supplementary tables), though, that is not about mediation but is about the associations between SES (at t1) and job strain and ERI (at t2) and onset of depressive disorder (after t2), analyses should NOT be adjusted for job strain at t1. This is because you want to show in table 2 the association between job strain/ERI at t2 and risk of depressive disorder after t2. If you adjust for job strain at t1, then you are showing estimates for the change of job strain from t1 to t2 and risk of depressive disorder after t2 and this is, obviously, a different topic. So, please do not adjust the estimates in table 1 for job strain at t1.

We agree entirely with your reasoning. The analyses shown in Table 1 were indeed not adjusted for job strain at T1. We have now repeated the analyses in the corresponding Suppl. Table 4 in the same manner, without adjusting for job strain at T1. The legends now reflect the actual analyses.

6) However, the estimates for job strain and ERI in table 1 should be adjusted for SES. Because SES can easily be a confounder for the association between job strain/ERI and risk of depressive disorder. If you do not want to adjust the estimates for job strain/ERI for SES in table 1, then please do this in a table in the supplementary analysis, but please make sure that there is a table somewhere in the manuscript that shows the estimates for job strain/ERI adjusted for at least sex, age and SES (and other sociodemographic variables and health behaviours, if you want). This will also allow other researchers to include your estimates in future meta-analyses.

Thank you for this suggestion. The estimates for job strain and ERI in Table 1 had indeed already been adjusted for SES indicators at T1, and the legend now reflects this. We have repeated the complete case analyses for Suppl. Table 4 in the same manner (adjusting for sex, age, SES indicators and other covariables), and this legend now also reflects this.

7) Table 1: The estimates for the SES measures are adjusted for age and sex, whereas the estimates for job strain/ERI are adjusted for age, sex, marital status, presence of children in the household, smoking, alcohol consumption and physical activity). What is the rational for adjusting SES and job strain/ERI for different sets of covariates?

Although education level is, in principle, a variable with a causal contrast, as the level of education can be manipulated in a real or imaginary experiment, it builds up monotonously, usually arriving at its final value at the beginning of adult life (2). All variables that we have measured (except sex and age) may appear later in life than education level. Therefore, the association of education with depression should not be adjusted for these variables, which would constitute possible mediators.

The PROQ cohort consists of participants with workplace stability, and therefore the same reasoning also applies, within limits, to the other two socioeconomic indicators (family income and type of occupation): they were measured at T1, but often established much earlier, and their association with depression should not be adjusted for any variables that might have arisen later.

On the other hand, as suggested by the reviewer in comment 6, the estimates for job strain and ERI at T2 should be adjusted for SES at T1, and we have done so. Following the same reasoning, we have also adjusted for other potential confounders. In parallel, following STROBE guidelines and since we wanted to know if the inclusion of these other potential confounders might affect the estimated effects of job strain and ERI on depression, we also performed a complete cases analysis adjusted only for age and sex (model 1 in Suppl. Table 4). These estimates do not differ strongly from the more adjusted estimates for complete cases (revised model 2 in Suppl. Table 4).

8) All tables: In the legends you write: “and for lifestyle habits at t1 and t2 (marital status, presence of children in the household, smoking, alcohol consumption and physical activity)”. Sounds a little bit strange to call “marital status” and “presence of children in the household” a “lifestyle habit”. Please consider to revise. Please also clarify if “physical activity” means “leisure time physical activity”.

We agree with the reviewer's suggestion to reconsider the terminology used in the legends. Referring to "marital status" and "presence of children in the household" as "lifestyle habits" might indeed sound a bit strange. We have amended the manuscript using clearer terminologies (p. 7, ll. 143s. and legends).

9) Table 1: Please provide the following information in the table: Number of participants in each exposure group, number of cases in each exposure group, number of cases per 1000 person-years in each exposure group.

Thank you for this suggestion. The numbers of exposed participants and of cases can only be given for a complete cases analysis. Our main results (shown in Table 1) use sequential multiple imputation for those participants that did not e.g. respond to questions about SES and about psychosocial stress at work (see also S1 Fig). The imputed answers will be slightly different in each imputed data set. We have therefore addressed your request by amending S3-S4 Tables, which contain the results for the complete cases of the cohort.

10) I am missing a table showing the association between the exposure and the mediator. For example, correlations between SES at t1 and job strain at t1, SES at t1 and job strain and ERI at t2, SES at t2 and job strain and ERI at t2. I am also missing an analysis showing whether SES at t1 predicts job strain at t2 (after adjustment for job strain at t1). And maybe also an analysis whether SES at t1 predicts ERI at t2 (after adjustment for job strain at t1). Thus, some data showing that the assumption that exposure to low SES is associated with an increased risk of job stress is confirmed in this study.

Thank you for this suggestion. We have added a table showing the prospective association between SES at T1 and both ERI and job strain at T2, adjusted for job strain at T1 (S5 Table, discussed on p. 13, ll. 256-257). We chose to add these prospective associations, instead of cross-sectional correlations, since these associations are those that are incorporated in our mediation model.

We agree with you that the cross-sectional relationship between SES and job strain in the same data collection wave can only be considered a correlation, not a causal effect. Since the manuscript attempts to estimate causal effects from our prospective cohort, we do not consider such cross-sectional correlations to be suitable for inclusion in the MS. However, we have attached a version of this table including the cross-sectional correlation to the cover letter, as it might nevertheless be of interest to you (Suppl_SES_PSW_Reviewer1.docx).

Minor comments

11) Introduction, page 3, lines 58-60: “These stressors, in turn, are associated with the onset of several physical and mental health problem (note: a plural “s” is missing here) (references 10-12), including depression (references 13,14). Reference #24 (Rugulies et al 2017) that appear later in manuscript, could be added here to references #13 and #14. Reference #11 (Mikkelsen et al) would fit better with references #13 and #14 than with references #10 and #12. Consider to add to references #10 and #12, a reference to a recent meta-review on job stressors and various health outcomes by Niedhammer et al 2021 (https://www.ncbi.nlm.nih.gov/pubmed/34042163)

Thank you for this observation. We have amended the manuscript (p. 3, ll. 60-61).

12) Methods, page 6, lines 116-118: “Imbalance was considered present when the sum of scores for psychological job demands, a proxy of the Siegrist effort scale (reference 21) was higher than that for reward.” Is this equal to calculating a ratio between efforts (or in this case job demands) and rewards and then using a ratio >1.0 as an indication of ERI? If this is equal then please mention this, because this would mean that you have calculated ERI in a similar way as in several previous studies.

Thank you for the opportunity to clarify. While the two formulations are indeed mathematically equivalent, we have now adopted the description more common in the literature (p. 6, ll. 118-121).

13) Results page 10, line 221-222: “but SES indicators were more strongly related to its incidence in men (Fig. 1A-C)”. Should this be “Fig. 1A-D” instead?

Thank you for your suggestion, which we have adopted (p. 11, ll. 224).

14) Results page 11, line 232: “investigated separately for men (Table 2), for women (Table 3) and for both together (Table 4)”. I suggest to write “for both sexes together (Table 4)”.

Thank you for your suggestion, which we have adopted (p. 13, l. 261).

15) Results page 11, at several places (and maybe also at other places in the paper): There is the abbreviation “SSE”. What is this? Or is this a typo and should mean SES?

Thank you for this observation (SSE is derived from the French term). We have amended the manuscript to read « SES » everywhere.

16) Discussion, page 15, lines 303-307: “Regarding causal mechanisms…”This is a very long sentence that is not easy to understand. Please break down into two sentences.

Thank you for this suggestion. We have broken down this sentence into three (p. 19, ll. 353-357).

17) Discussion, page 18, line 370-373: “For example, the 2021 Noble prize in economy…”. Please add a reference to the study that demonstrated that increasing the minimum wage had no negative effect on employment.

and

18) Discussion, page 18, line 373-374: “Regarding psychosocial stressors at work, countries such as Iceland have decreased weekly work hours”. First, if you want to keep this sentence, then please add a reference. Second, I am not sure about the relevance here. In your paper you did not define job stressors by lengths of working hours but by other parameters, so I am wondering about the relevance of this sentence. I suggest that you consider to delete this sentence about working hours in Iceland.

Thank you for the suggestions. While there are indeed studies for the minimum wage effect (3), we have followed the recommendation of Reviewer 2 (comment 13) and have now limited the paragraph to the conclusion of our study (p.22).

################################################################

Reviewer #2

Methods:

1) Line 99: On what basis were the household income cutoffs chosen?

We have now added an explanation of the cutoffs in the manuscript (p. 5, ll. 99-102): household income was categorized as less than CAD40 000, CAD40 000 to CAD69 999, or at least CAD70 000 before taxes, taking into account the mean national household income of CAD 44,783.

2) Lines 108-114: Please give more information on the used instruments, i.e. range of values, median of the representative sample of Quebec workers.

We have provided additional information on the used instruments in the manuscript (p. 6, ll. 114-116).

3) Lines 115-118: Was the effort part of the ERI not assessed? Please explain. If you have data on effort, please also provide these results and the known ERI.

Thank you for the suggestion. We have now amended the manuscript (p.6, ll. 118-119): “Since the Siegrist effort scale was not measured in our cohort, we used psychological job demands as a proxy (24).”

4) Lines 128-132: It is not clear to me what happened to individuals who received a depression diagnosis more than a year before T2.

Please refer to our response to comment 2 from reviewer 1. We have included a supplementary analysis in which we excluded all participants diagnosed with depression between 1991-01-01 and T2 (new S9-S11 Tables, discussed on p. 10, ll. 207-209 and p. 16, ll. 296-297).

5) Line 132: I assume that the second wave and the following wave are different names for the same thing. Is that correct? If yes, a consistent designation would be helpful. If no, please make the difference clearer.

We have now uniformly employed the term T2 throughout the manuscript to refer to the collection of cohort questionnaires in 1999-2001 The follow-up is the period of three years following T2.

6) Lines 153ff: I miss an introduction to statistical analysis. It should be mentioned that a time series analysis is performed with the dependent variable "depression yes/no". Is this correct? Readers with no prior experience in mediation analysis should be better considered.

Thank you very much for this suggestion. We have added a brief introduction that summarizes the main objective of developing our mediation method (p.8, ll. 156-157)

7) Lines 155+156: What is a natural direct effect or natural indirect effect? I don't understand the "natural".

In mediation analysis, we can decompose the total effect of low SSE on the incidence of depression into the natural direct effect, which quantifies the effect of SSE on depression independently of any pathways through which PSW operate, and the natural indirect effect, which necessarily depends on a change in the level of PSW caused by SSE and consequently leads to depression.

The term "natural" refers to the distribution of mediator levels in the population. Here, PSW is not fixed at an artificial or arbitrary value (such as the complete absence of PSW in the population). Instead, we allow the mediator variable to take the value found for each individual at the corresponding level they would naturally have if they were not exposed to SSE.

We have now added a reference about the concept of natural direct and indirect effects (p. 8, ll.159-163)

Results:

8) How relevant is the analysis of occupation type? Occupation type is very broad and describes very heterogeneous groups. What conclusions can be drawn from such a variable? Since these results are neither described nor discussed, I wonder if these analyses are relevant and should be presented in the main body of the paper.

An individual’s occupational class provides not only an indication of social prestige and the functional consequences of education and social connections, but also a direct reflection of the circumstances relevant to health through the physical and psychologic environment of the workplace itself (2, 4).

In the current study, we stress that the inequalities between men and women are most pronounced for occupational type (p. 10, ll. 216-218). This may indicate that women have to surmount additional obstacles to obtain a high occupational position, even when they have the same educational level as men. We also discuss the results obtained for occupational type on ll. 224-227; 243-245, 256-257; 287-288; 301-302 and 326-327.

9) Tables S2: A total column per sex would be convenient in Table S2. A subdivision of S2 into S2a (stratification by education), S2b (stratification by household income), etc. could improve clarity.

Thank you for this suggestion. We have amended the S2 Table.

10) Line 207: I cannot find ERI in Table S2.

The Suppl. Table 2 reference to cohort characteristics at baseline (1991-1993). Unfortunately, we do not have a measure of ERI at baseline because that instrument had not yet been published. We have, however, now included numbers of participants exposed to job strain and to ERI at T2 (1999-2003) in S4 Table.

11) Figure 1: Please check if in figure 1B (household income) the graphs for the sexes are not reversed. Because in the tables you see the negative impact of low household income for men and not for women.

Thank you for this suggestion. We have now double-checked the graphs, and they are indeed correct. The negative impact of low household income can be seen in the spread between the blue and red lines.

12) Lines 219-220: I struggle with your sentence „In women, results were inconclusive: the confidence intervals include both a considerable 220 protection and an increased risk of depression at the lower levels of SES indicators.“.

Isn't it more likely that SES may not have a statistically significant impact on a depression diagnosis in women?

Thank you for this suggestion. We have improved this sentence in the manuscript (p.12, ll. 246-247).

13) Conlcusion: In my opinion, the comments on the 2021 Nobel Prize and iceland are not conclusions. Here, only conclusions of the study should be mentioned and no further discussion should be led.

We have now restricted the paragraph to conclusions of our study (p.22).

Minor remarks:

Please remember, abbreviations need to be introduced only once and should be used after their introduction (e.g. ERI or SES).

Thank you for calling this out. We have now used abbreviations everywhere after their introduction, except in section titles and table titles where we have preferred to use the full expressions.

Line 88: Please explain the abbreviation CHU.

Figure 1: Typo: It should be „sex“

Typo: SSE in several lines.

Thanks, we have fixed all of these. 

Line 143: Please check your information on alcohol consumption. In Table S2, for example, 6 doses/week was used as the cut-off and here 5 drinks are mentioned.

Thanks for pointing out the inconsistent terminology. We have changed “>5 drinks” to the mathematically equivalent “≥ 6 drinks” at line 145-146 to be consistent with S2 Table.

Line 276: Please provide the percentage of women with the highest level of education and do not just refer to Table S2.

We have added this information (p. 16, l. 327).

Line 300: Typo 95%IC

We have amended the manuscript (p.19, l. 350).

References 

1. Pena-Gralle APB, Talbot D, Trudel X, Aubé K, Lesage A, Lauzier S, et al. Validation of case definitions of depression derived from administrative data against the CIDI-SF as reference standard: results from the PROspective Québec (PROQ) study. BMC psychiatry. 2021;21(1):491.

2. Lash TL, VanderWeele TJ, Haneuse S, Rothman KJ. Social Epidemiology. Modern Epidemiology. 4th ed. ed. Philadelphia: Wolters-Kluwer; 2021. p. 1005-28.

3. Card D, Krueger AB. Minimum wages and employment: A case study of the fast food industry in New Jersey and Pennsylvania. American Economic Review. 1994;84(4):772-93.

4. Marmot M, Allen J, Bell R, Bloomer E, Goldblatt P. WHO European review of social determinants of health and the health divide. Lancet (London, England). 2012;380(9846):1011-29.

---

## [Decision Letter · Decision Letter 1]

15 Aug 2023

PONE-D-23-03329R1Socioeconomic inequalities, psychosocial stressors at work and physician-diagnosed depression: time-to-event mediation analysis in the presence of time-varying confoundersPLOS ONE

Dear Dr. Pena-Gralle,

Thank you for submitting your manuscript to PLOS ONE. After careful consideration, we feel that it has merit but does not fully meet PLOS ONE’s publication criteria as it currently stands. Therefore, we invite you to submit a revised version of the manuscript that addresses the points raised during the review process.

Reviewer 1 was again willing to evaluate your revised submission and has some comments to your resubmission. Reviewer 2 feels that all the questions listed have been adequately answered. I would like to ask what are the differences between "direct and indirect effects" and "natural direct and natural indirect effects" (see lines 156-160). While reading, I noticed the French abbreviation "SSE" in Figure 1D.

Please address the comments when revising the manuscript and please submit a clean version and a revised version. The yellow markings as in your first revision are not very convenient and especially do not indicate which sentences/words have been deleted in individual places. Therefore, please use the revision mode for resubmission.

We look forward to receiving your revised manuscript.

Kind regards,

Swaantje Wiarda Casjens

Academic Editor

PLOS ONE

Journal Requirements:

Reviewers' comments:

Reviewer's Responses to Questions

**Comments to the Author**

1. If the authors have adequately addressed your comments raised in a previous round of review and you feel that this manuscript is now acceptable for publication, you may indicate that here to bypass the “Comments to the Author” section, enter your conflict of interest statement in the “Confidential to Editor” section, and submit your "Accept" recommendation.

Reviewer #1: (No Response)

2. Is the manuscript technically sound, and do the data support the conclusions?

Reviewer #1: Yes

3. Has the statistical analysis been performed appropriately and rigorously? 

Reviewer #1: Yes

4. Have the authors made all data underlying the findings in their manuscript fully available?

Reviewer #1: No

5. Is the manuscript presented in an intelligible fashion and written in standard English?

Reviewer #1: Yes

6. Review Comments to the Author

Reviewer #1: Thank you for your revision and for addressing my concerns. Most of my concerns and comments have successfully been addressed. I have the following remaining points that I would like to ask you to address in another revision.

1) You explained in the response letter that all of your analyses were based on ICD-9 codes and that you used the ICD-9 codes 296.x, 300.4 and 311.x. 296.x includes codes on Affective Psychoses, thus, this is a mixture of what we call today unipolar depressive disorder and bipolar disorder. 300.4 is neurotic depression, 311 is depressive disorder, not otherwise classified. I was wondering, why you listed 311.x instead of 311, as far as I know there is only one 311 code in the ICD-9 (or maybe you did use the ICD-9 CM, clinical modification?). That the codes for 296.x include both codes for depressive and manic/bipolar disorder needs to be addressed in the paper. Either by limiting your analyses to those codes that are likely depressive disorder codes (i.e. re-running your analyses while excluding the manic/bipolar disorder codes) or by addressing in the discussion section, how certain you can feel that your study was about depressive disorders as opposed to be about affective disorders. It would also be helpful, if you list in the appendix all codes that you included (i.e. all subgroup codes from 296.x) and if you can make clear if this was ICD-9 or ICD-9 CM.

2) Supplementary table 5 shows the association between SES and a) job strain (at t2 and at t2 adjusted for job strain at t1) and b) ERI (at t2 and at t2 adjusted for job strain at t1). There are some interesting results here: Education is not related to job strain, income is related to job strain (lower income, more job strain), occupational grade is related to job strain (lower grade, more job strain). So, here the associations are in the direction that you assumed, lower SES is associated with more psychosocial work stressors (job strain).

It is the opposite, though, when psychosocial work stressors are measured by ERI instead of job strain. Lower education is associated with less ERI and lower occupational grade is associated with less ERI (no association for income and ERI). Thus, the basic assumption of your mediation analysis, that the exposure, low SES, is associated with higher levels of the mediator, psychosocial stressors, is confirmed only for job strain but not for ERI, where the association is the opposite as originally assumed. You briefly address these findings in the result section, but you do not discuss them thoroughly in the discussion section.

I would like to ask you to do two things: First, to repeat the analyses in Table S5 for men and women. This seems to be prudent, as your main analyses are also sex-stratified. Would be interesting to see, if the pattern of association between SES-Job Strain and SES-ERI are similar or different in men and women.

Second, please discuss the results on the association between the exposure and the mediator in the discussion section more thoroughly. Why are individuals of low SES more likely to experience job strain but less likely to experience ERI? Isn’t this quite surprising, considering that job strain and ERI conceptually overlap (both are measures of psychosocial stressors at work)? And in this study here, job strain and ERI even overlap in the measurement, since the psychological demand component in the job strain measure and the effort component in the ERI measure are based on the same items. But still, the association with SES goes into different directions of job strain and ERI? What could be explanations for this?

7. PLOS authors have the option to publish the peer review history of their article (what does this mean?). If published, this will include your full peer review and any attached files.

Reviewer #1: No

---

## [Author Response · Author response to Decision Letter 1]

3 Oct 2023

Please see the attached cover letter with our responses to the reviewers' comments.

---

## [Editor Report · Decision Letter 2]

12 Oct 2023

Socioeconomic inequalities, psychosocial stressors at work and physician-diagnosed depression: time-to-event mediation analysis in the presence of time-varying confounders

PONE-D-23-03329R2

Dear Dr. Pena-Gralle,

We’re pleased to inform you that your manuscript has been judged scientifically suitable for publication and will be formally accepted for publication once it meets all outstanding technical requirements.

Kind regards,

Swaantje Wiarda Casjens

Academic Editor

PLOS ONE

Additional Editor Comments (optional):

The answers to the outstanding questions have been processed and answered satisfactorily.

However, there are duplicate versions of the supporting tables in the PlosOne system, which needs to be addressed. Also, table numbers and headings would be helpful with the supporting tables.

---

## [Editor Report · Acceptance letter]

16 Oct 2023

PONE-D-23-03329R2 

Socioeconomic inequalities, psychosocial stressors at work and physician-diagnosed depression: time-to-event mediation analysis in the presence of time-varying confounders 

Dear Dr. Pena-Gralle:

I'm pleased to inform you that your manuscript has been deemed suitable for publication in PLOS ONE. Congratulations! Your manuscript is now with our production department. 

Kind regards, 

on behalf of

Dr. Swaantje Wiarda Casjens 

Academic Editor

PLOS ONE